# Assay System for Simultaneous Detection of SARS-CoV-2 and Other Respiratory Viruses

**DOI:** 10.3390/diagnostics11061084

**Published:** 2021-06-13

**Authors:** Ho-Jae Lim, Jung-Eun Park, Min-Young Park, Joo-Hwan Baek, Sunkyung Jung, Nackmoon Sung, Jae-Hyun Yang, Min-Woo Lee, Sun-Hwa Lee, Yong-Jin Yang

**Affiliations:** 1Department of Molecular Diagnostics, Seegene Medical Foundation, Seoul 04805, Korea; 52rotc.hjl@mf.seegene.com (H.-J.L.); pyli186@mf.seegene.com (M.-Y.P.); L15057@mf.seegene.com (J.-H.B.); skjung@mf.seegene.com (S.J.); lshkim@mf.seegene.com (S.-H.L.); 2Department of Integrative Biological Sciences & BK21 FOUR Educational Research Group for Age-Associated Disorder Control Technology, Chosun University, Gwangju 61452, Korea; jepark@chosun.ac.kr; 3Clinical Research Institute, Seegene Medical Foundation, Seoul 04805, Korea; paratb@gmail.com; 4Paul F. Glenn Center for Biology of Aging Research, Department of Genetics, Blavatnik Institute, Harvard Medical School, Boston, MA 02115, USA; Jae-Hyun_Yang@hms.harvard.edu; 5Department of Integrated Biomedical Science and Soonchunhyang Institute of Medi-Bio Science (SIMS), Soonchunhyang University, Cheonan-si 31151, Korea; mwlee12@sch.ac.kr

**Keywords:** SARS-CoV-2, influenza virus, respiratory syncytial virus, simultaneous detection

## Abstract

Severe acute respiratory syndrome coronavirus 2 (SARS-CoV-2) triggers disease with nonspecific symptoms that overlap those of infections caused by other seasonal respiratory viruses (RVs), such as the influenza virus (Flu) or respiratory syncytial virus (RSV). A molecular assay for accurate and rapid detection of RV and SARS-CoV-2 is crucial to manage these infections. Here, we compared the analytical performance and clinical reliability of Allplex™ SARS-CoV-2/FluA/FluB/RSV (SC2FabR; Seegene Inc., Seoul, South Korea) kit with those of four commercially available RV detection kits. Upon testing five target viral strains (SARS-CoV-2, FluA, FluB, RSV A, and RSV B), the analytical performance of SC2FabR was similar to that of the other kits, with no significant difference (*p* ≥ 0.78) in z-scores. The efficiency of SC2FabR (E-value, 81–104%) enabled reliable SARS-CoV-2 and seasonal RV detection in 888 nasopharyngeal swab specimens processed using a fully automated nucleic acid extraction platform. Bland–Altman analyses revealed an agreement value of 95.4% (SD ± 1.96) for the kits, indicating statistically similar results for all five. In conclusion, SC2FabR is a rapid and accurate diagnostic tool for both SARS-CoV-2 and seasonal RV detection, allowing for high-throughput RV analysis with efficiency comparable to that of commercially available kits. This can be used to help manage respiratory infections in patients during and after the coronavirus disease 2019 pandemic.

## 1. Introduction

Viruses are the most common cause of respiratory infections, which often lead to comorbidity and mortality in children and adults [1,2]. The early detection of respiratory viruses (RVs) is essential for reducing the risks associated with infection, nosocomial transmission, and inappropriate treatment [3,4]. Severe acute respiratory syndrome coronavirus 2 (SARS-CoV-2), a positive-sense single-stranded RNA virus, causes coronavirus disease 2019 (COVID-19) [5,6]. This virus emerged in the human population in the final months of 2019 and then spread worldwide, with 105 million confirmed cases and approximately 2.3 million deaths as of 9 February 2021 [7].

SARS-CoV-2, influenza virus (Flu), and respiratory syncytial virus (RSV) are major pathogens that primarily target the human respiratory system [8,9]. SARS-CoV-2 causes fairly nonspecific symptoms, and the onset of disease coincides with the active Flu and RSV season [10,11]. The non-specific symptoms and scarce clinical knowledge of the SARS-CoV-2 infection can mislead patient diagnoses [12]. Considering the difficulty in diagnosing simultaneous infection with SARS-CoV-2 and Flu A, it is currently difficult to analyze the overlapping clinical symptoms or frequency of these two viral infections [13]. Approximately 3% of RV infections occur simultaneously with SARS-CoV-2 infection, with RSV being the most common, followed by the Flu [14]. Since co-infection leads to more severe symptoms than infection caused by a single virus [15], a pressing clinical need exists for broad and accurate tools to detect co-infection [16,17].

The US Food and Drug Administration has approved some commercial kits for emergency use for coronavirus detection [18]. Further, the Allplex™ respiratory panel 1 (RP1) and Advansure™ RV-plus (RV-plus) kits are well established for Flu and RSV detection, respectively [19,20]. However, the option for the simultaneous detection of all three viruses is limited. Recently, the Allplex™ SARS-CoV-2/FluA/FluB/RSV (SC2FabR) assay was released on the market for the simultaneous detection of SARS-CoV-2, Flu, and RSV. In the current study, we compared the efficiency and molecular diagnostic performance of SC2FabR for the detection of simultaneous RV co-infection with those of four commercially available kits, the Allplex™ 2019-nCoV kit, Standard M n-CoV Real-Time Detection kit, Allplex™ Respiratory panel 1 kit, and Advansure™ RV-plus Real-Time RT-PCR kit, using 870 clinical samples.

## 2. Materials and Methods

### 2.1. Clinical Specimen Collection and Storage

Anonymized residual nasopharyngeal swab (NPS) specimens (888) were obtained and preserved from February to September 2020, as part of the routine procedure for SARS-CoV-2 testing, and from March 2017 and December 2019, for seasonal RV testing for the presence of Flu A, Flu B, RSV A, or RSV B. All procedures were approved by the institutional review boards at the Seegene Medical Foundation (SMF-IRB-2020-012). Among the NPS specimens, 380 were classified as positive or negative for SARS-CoV-2, and 490 were classified as positive or negative for Flu A, Flu B, RSV A, or RSV B. Additionally, 18 samples were used to evaluate clinical performance of SC2FabR for simultaneous detection of co-infection. All samples were processed using an automated nucleic acid extraction system, MagNA PURE 96 (Roche, Basel, Switzerland), according to the manufacturer’s protocol [21]. Nucleic acids were extracted from 200 μL of each specimen. Nucleic acids were eluted in 100 μL elution buffer and stored at −80 °C until use.

### 2.2. Commercial Assay Kits for RV Detection

SARS-CoV-2, Flu, and RSV were simultaneously detected using the Allplex™ SARS-CoV-2/FluA/FluB/RSV (SC2FabR) kit (Seegene Inc., Seoul, South Korea). The Allplex™ 2019-nCoV kit (Seegene Inc., Seoul, South Korea) was used for the detection of genes encoding the SARS-CoV-2 envelope protein (E), RNA-dependent RNA polymerase (RdRp), and nucleocapsid protein (N). The Standard M n-CoV Real-Time Detection kit (SD BIOSENSOR Inc., Suwon, South Korea) was used for the detection of *E* and *RdRp* genes of SARS-CoV-2. The Allplex™ Respiratory Panel 1 kit (RP1; Seegene Inc., Seoul, Korea) was used for the detection of Flu A (subtype H1, H1 pdm09, and H3), Flu B, RSV A, and RSV B. The Advansure™ RV-plus real-time RT-PCR kit (RV-plus; LG Chem Ltd., Seoul, South Korea) was used for the detection of Flu A, Flu B, RSV A, RSV B, human enterovirus, bocavirus, and human coronavirus OC43. All assays were performed following the manufacturers’ protocols. The main characteristics of the kits compared in the current study, such as the manufacturer, release year, specimens, platform, running time, target information, maximum capacity for 96-well plate, and turnaround time, are shown in Table 1. Any discrepancies in test outcomes were further assessed using the Allplex™ II RV16 Detection kit (RV-16; Seegene Inc., Seoul, South Korea). The test results were regarded as true positives if more than two assays gave a positive reading.

### 2.3. Analytical Performance of Commercial Kits

The following RV strains were used for determining the analytical sensitivity of the assayed kits: SARS-CoV-2 (KRISS 111-10-506), from the Korea Research Institute of Standards and Science (KRISS; Daejeon, South Korea); (NCCP 43381-43387), from National Culture Collection for pathogens (NCCP; Cheongju, South Korea); and Twist Bioscience (79683; San Francisco, CA, USA); Flu A (VR-810), Flu B (VR-1735), RSV A (VR-41) and RSV B (VR-955), from the American Type Culture collection (ATCC; Manassas, VA, USA). As SARS-CoV-2 from KRISS does not harbor the gene encoding the spike protein, PCR efficiency was determined using a specimen positive for this gene. SARS-CoV-2 particles from the original sample were diluted to 7 × 10^4^ copies/μL, and the concentrations of the Flu and RSV strains were set to 0.5 ng/μL. The samples were then serially diluted 10-fold from 10^4^ to 1 for PCR optimization.

### 2.4. Statistical Analysis

All statistical analyses were performed using SPSS (v25.0, IBM Corp, Armonk, NY, USA) for Windows. Density plots were generated using ggplot2 packages in R (v4.1.0, R studio Inc., Boston, MA, USA). The kit performance efficiency was calculated from the linear regression slope using the following formula: E value = 100 × (−1 + 10^−1/slope^) [5]. Raw cycle threshold (Ct) values for each experiment were transformed to z-scores for performance evaluation. The z-scores were calculated by subtracting the mean positive Ct values from the raw Ct values and then dividing this by the standard deviation (SD) of positive Ct values, according to the following formula: z-score = (Ct value − mean Ct value)/SD [22]. When three assays yielded a consistent result, that result was considered likely true. The statistical significance of z-score differences was calculated using a Student’s *t*-test or the analysis of variance with a Bonferroni post-hoc test for two groups or three groups, respectively. Bland–Altman analysis was performed for positive matched datasets for each variable. The presence of proportional bias was calculated by testing the slope of the regression line fitted to the Bland–Altman plot [23].

## 3. Results

### 3.1. Comparison of PCR Efficiency of the Virus Detection Systems

We evaluated the PCR efficiency of each target assay by analyzing 10-fold serial dilutions of viral RNA samples of SARS-CoV-2 (Figure 1), as well as Flu and RSV (Figure 2). As shown in Figure 1 and Figure 2, the E and R^2^ values of the five analyses were >80% and 0.991, respectively. To evaluate the test performance, we then calculated the z-sores from the mean values of normalized Ct values (Figure 3 and Figure 4). No significant differences (*p* ≥ 0.78) in the z-scores between the analysis systems were apparent (Figure 3 and Figure 4), indicating that the PCR efficiency of the SC2FabR kit is high and similar to that of the commercially available SARS-CoV-2 and RV analysis systems.

### 3.2. RV Distribution

We analyzed 870 NPS specimens in the current study. Of these, 380 specimens were classified using the 2019-nCoV assay as SARS-CoV-2-positive (180) or SARS-CoV-2-negative (200). The remaining 490 specimens were sorted based on the RP1 assay as RV-positive or RV-negative. For the former, 265 specimens were positive for at least one respiratory virus; 251 were positive for one virus and 14 were positive for two viruses.

### 3.3. Clinical Performance of the Detection Systems

As shown in Table 2, based on the SC2FabR assay, 180 of 380 NPS samples were positive for SARS-CoV-2. The sensitivity and specificity of the SC2FabR assay were as high as those of the 2019-nCoV and the Standard M assays, with the ability to distinguish samples that were positive or negative for SARS-CoV-2, consistent with those of the two assays (Table 2). Following the clinical testing of 490 NPS samples for the presence of Flu and RSV, the SC2FabR assay identified 99, 91, and 75 samples as positive for Flu A, Flu B, and RSV, respectively. Compared with that of the RP1 assay, the sensitivity of the SC2FabR assay was 100% (99/99) for Flu A, 100% (91/91) for Flu B, and 98.7% (74/75) for RSV, with 100% specificity for all targets. Compared with that of the RV-plus assay, the sensitivity of the SC2FabR assay was 99.0% (98/99) for Flu A, 100% (91/91) for Flu B, and 92.0% (69/75) for RSV, and the specificity levels were 99.5% for Flu A and RSV and 99.7% for Flu B (Table 2). The diagnostic accuracy of the SC2FabR assay was very high, at > 98.4%, compared with those of the four other detection systems. These results indicated that the SC2FabR kit can be used to simultaneously detect SARS-CoV-2, Flu, and RSV, with very high accuracy.

To evaluate the error rate of the SC2FabR kit, we analyzed 13 discrepant samples using the RP1, RV-plus, and RV-16 analysis systems and compared the results with those obtained with the SC2FabR kit (Table 3). The SC2FabR test results for 11 samples were confirmed and every sample was assigned following SC2FabR results (Table 3). These observations suggest that the error rate of SC2FabR for SARS-CoV-2, Flu, and RSV is equal to zero.

### 3.4. Intersystem Comparison of the z-Scores Calculated from the Ct Values

We then performed the Bland–Altman analysis to assess the level of agreement of test results obtained with the SC2FabR and 2019-nCoV kits (Figure 5A,C) or the SC2FabR and Standard M kits (Figure 5B). An analysis of 179 SARS-CoV-2-positive clinical specimens demonstrated an exceptional degree of agreement between the SC2FabR and 2019-nCoV assays, specifically 96.1% (172/179) consistency for *RdRp* gene detection and 96.6% (173/179) consistency for *N* gene detection. Likewise, a comparison with the Standard M data revealed 94.9% (170/179) consistency for *RdRp* gene detection. Further, testing of 259 Flu- or RSV-positive clinical specimens demonstrated a comparable level of agreement between the SC2FabR and RP1 assays, with 93.9% (92/98), 92.3% (84/91), 97.1% (33/34), and 94.4% (34/36) consistency for Flu A, Flu B, RSV A, and RSV B detection, respectively (Figure 5D–G). Furthermore, the SC2FabR and RV-plus assays showed 96.9% (95/98), 96.7% (88/91), 91.2% (31/34), and 94.4% (34/36) consistency for Flu A, Flu B, RSV A, and RSV B detection, respectively (Figure 5H–K). More than 95.4% of samples tested using the different assays fell between the set SD boundaries, indicating an exceptionally high correlation between the SC2FabR data and those of the well-established assays.

## 4. Discussion

The global coronavirus pandemic has been affecting the human population since 2019 [24]. The disease is transmitted via aerosols during close unprotected contact with infected individuals [25]. Great efforts are continuously being devoted to help contain viral respiratory infections by developing rapid and accurate infection diagnosis and analysis systems [17,26]. The rapid, broad, and accurate diagnosis of the SARS-CoV-2 infection is crucial for effective management and control of the spread of the disease in the population [16,27].

In the current study, we compared the analytical performance of the SC2FabR platform with that of four other commercial analytical platforms. The five platforms tested herein (SC2FabR, 2019-nCoV, Standard M, RP1, and RV-plus) showed high PCR efficiency (Figure 1, Figure 2, Figure 3 and Figure 4) and similar clinical performance (Table 2 and Figure 5) when used to detect SARS-CoV-2, Flu, and RSV. The analytical performance of the SC2FabR assay was consistent regardless of SARS-CoV-2 lineages (Appendix A). For SARS-CoV-2-containing samples, during the collection period, there was no issue regarding the SARS-CoV-2 variant in Korea. Therefore, the SARS-CoV-2 samples supposed to belong to the Wuhan-Hu-1 viral strain (NCBI accession number NC_045512.2). The clinical performance of the SC2FabR assay showed 100% diagnostic accuracy, and the assay results were consistent with those of the reference platforms (2019-nCoV or Standard M assays) (Table 2). In addition, the clinical performance of the SC2FabR kit for the analysis of seasonal RVs showed 98.4–100% diagnostic accuracy (Table 2), and the simultaneous detection of co-infection with SARS-CoV-2 presented 100% reliability for Flu A and RSV (Appendix A), which is similar to the results of previous studies [28]. Thirteen samples that yielded inconsistent results frequently gave “weakly positive” reactions in the SC2FabR assay, with the target amplification detected in the “gray” region [29], with Ct values of 36 to 40 (Figure 3 and Figure 4). Re-analysis using four other commercial kits revealed that every sample was assigned following SC2FabR results (Table 3). Several factors, such as target gene type, primers used, and assay sensitivity, might contribute to assay discrepancies [30,31]. A correlation of results of clinical evaluation across different platforms is the most important factor in disease diagnosis [32]. Therefore, in the current study, we converted the Ct values obtained using the five systems to z-scores to analyze the correlation between the platforms (Figure 5). We observed that the SC2FabR results showed a very high correlation (above 95.4%) with those of other platforms for SARS-CoV-2, Flu, and RSV analysis (Figure 5). These observations suggest that the SC2FabR kit can be employed as a useful analysis platform for the detection of simultaneous RV and SARS-CoV-2 infections. SC2FabR was also optimized using saliva samples that are collected non-invasively compared to NPS, which is indeed an advantage over other system. In summary, we here evaluated the accuracy, efficiency, and clinical utility of the SC2FabR assay. Simultaneous viral detection using the SC2FabR assay, comparable with that of other well-established assays, suggests the possibility of improving assay efficiency for the individual detection of SARS-CoV-2, Flu, and RSV viruses.

## 5. Conclusions

To our knowledge, this is the first study to compare the performance of the SC2FabR assay with that of four other well-established molecular assays. It is also the first study that reports normalized Ct value comparisons of z-scores for five molecular detection methods with a large number of samples analyzed. Taking into account the results of SC2FabR, we consider that it is high-throughput, labor- and cost-saving, and can be used to facilitate the simultaneous diagnosis of SARS-CoV-2 and other RVs during and even after the COVID-19 pandemic.

## Figures and Tables

**Figure 1 diagnostics-11-01084-f001:**
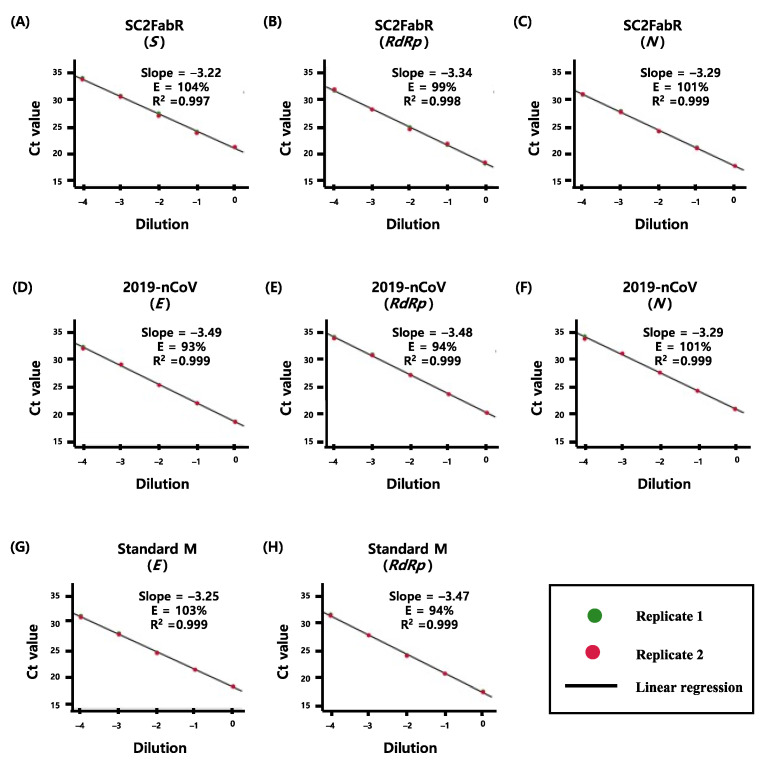
Comparison of the PCR efficiency of three assays [SC2FabR (**A**–**C**), 2019-nCoV (**D**–**F**), and Standard M (**G**,**H**)] for SARS-CoV-2 detection. The PCR efficiency (**E**) of the amplification of each target gene was assessed using duplicate 10-fold dilution series of SARS-CoV-2 viral RNA. Linear regression analysis was performed using the SPSS software to obtain the slope and R^2^ value. The percentage efficiency was calculated from the slope using the formula E = 100 × (−1 + 10^−1/slope^). *S*, gene encoding the spike protein of SARS-CoV-2; *E*, gene encoding the envelope protein of SARS-CoV-2; *RdRp*, gene encoding the RNA-dependent RNA polymerase of SARS-CoV-2; *N*, gene encoding the nucleocapsid protein of SARS-CoV-2; RT-PCR, reverse-transcription PCR; SARS-CoV-2, severe acute respiratory syndrome coronavirus 2.

**Figure 2 diagnostics-11-01084-f002:**
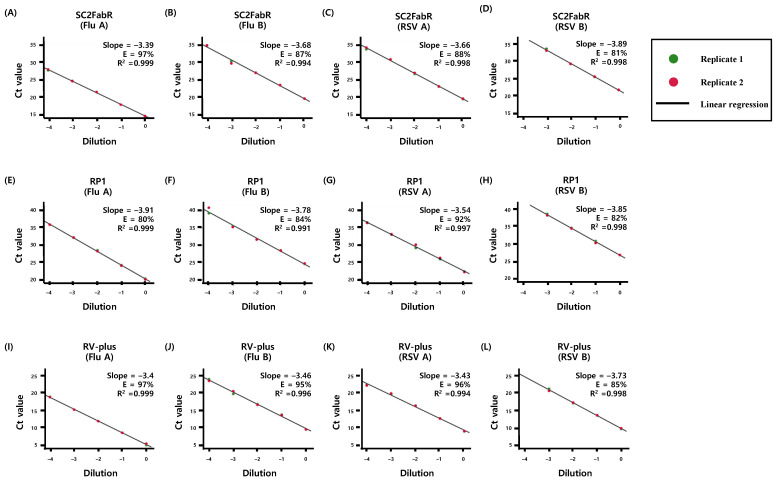
PCR efficiency for the detection of respiratory viruses. The PCR efficiency of the amplification of each target gene using three assays [SC2FabR (**A**–**D**), RP1 (**E**–**H**), and RV-plus (**I**–**L**)] was assessed using duplicate 10-fold dilution series of Flu and RSV viral RNA. Linear regression analysis was performed using the SPSS software to obtain the slope and R^2^ values. The percentage efficiency was calculated from the slope using the formula E = 100 × (−1 + 10^−1/slope^). Flu A, influenza virus type A; Flu B, influenza virus type B; RSV A, respiratory syncytial virus A; RSV B, respiratory syncytial virus B; RT-PCR, reverse-transcription PCR.

**Figure 3 diagnostics-11-01084-f003:**
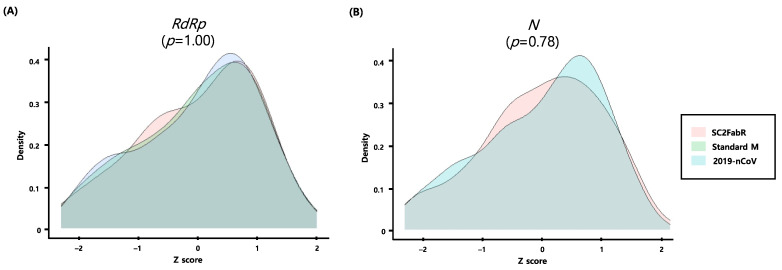
Densities of transformed Ct values obtained from SARS-CoV-2 data following within-assay normalization. Solid lines correspond to the SC2FabR, Standard M, and 2019-nCoV assays. *RdRp* gene, gene encoding the RNA-dependent RNA polymerase of SARS-CoV-2 (**A**); *N*, gene encoding the nucleocapsid protein of SARS-CoV-2 (**B**).

**Figure 4 diagnostics-11-01084-f004:**
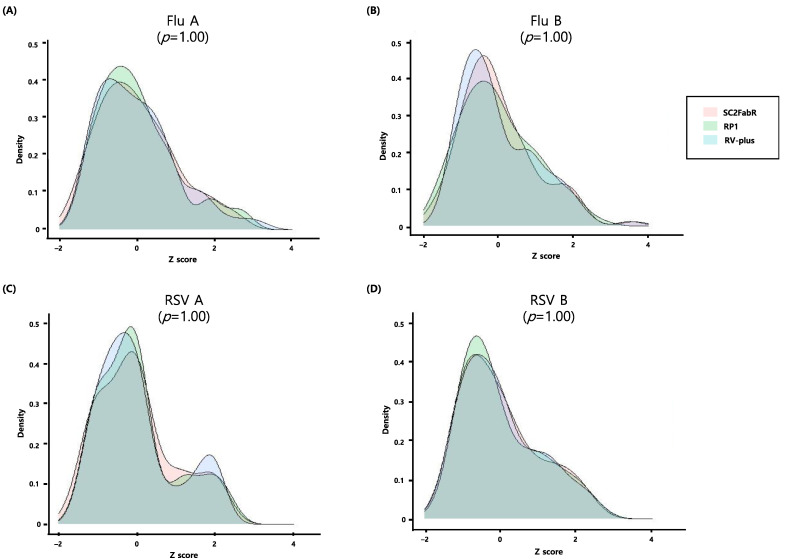
Densities of transformed Ct values obtained from SARS-CoV-2 data following within-assay normalization. Solid lines correspond to the SC2FabR, RP1, and RV-plus assays Flu A, influenza virus type A (**A**); Flu B, influenza virus type B (**B**); RSV A, respiratory syncytial virus A (**C**); RSV B, respiratory syncytial virus B (**D**).

**Figure 5 diagnostics-11-01084-f005:**
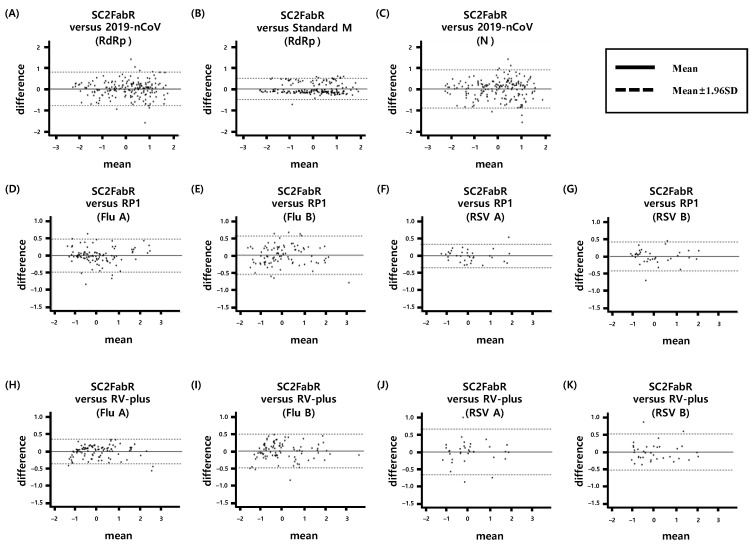
Bland-Altman analysis of the quantitative data from three molecular assays for SARS-CoV-2, Flu, and RSV. The analysis was performed using matched positive samples from all assays to compare the transformed Ct values from the SC2FabR and well-established assays. The mean Ct values are plotted on the x-axis, while the Ct difference between the two platforms for each sample are plotted on the y-axis. The mean and 1.96 SD border are shown. *RdRp* gene, gene encoding the RNA-dependent RNA polymerase of SARS-CoV-2 (**A**,**B**); *N*, gene encoding the nucleocapsid protein of SARS-CoV-2 (**C**); Flu A, influenza A virus (**D**,**H**); Flu B, influenza B virus (**E**,**I**); RSV A, respiratory syncytial virus A (**F**,**J**); RSV B, respiratory syncytial virus B (**G**,**K**).

**Table 1 diagnostics-11-01084-t001:** Overview of molecular in vitro diagnostic multiplex PCR assays used in this study.

Detection Kit (Release Year)	Manufacturer	Sample Type	Platform	Running Time	Target Regions of SARS-CoV2	Target of Respiratory Virus	Maximum Capacity (96 Well Plate)	TAT (76,000 Samples)
SC2FabR (2020)	Seegene, Inc.	NPS, NPA, BAL, TS, Sputum, Saliva	CFX96	~115 min	*S*, *RdRp*, *N*	Flu A, Flu B, RSV	94 samples	1551 hr
2019-nCoV (2020)	Seegene, Inc.	NPS, NPA, BAL, TS, Sputum	CFX96, ABI 7500	~110 min	*E*, *RdRp*, *N*	-	94 samples	1483 hr
STANDARD M (2020)	SD BIOSENSOR, Inc.	NPS, OS, Sputum	CFX96, ABI 7500	~98 min	*E*, *RdRp*	-	94 samples	1321 hr
RP1 (2016)	Seegene, Inc.	NPS, NPA, BAL	CFX96	~150 min	-	Flu A (H1, H1-pdm09, H3), Flu B, RSV A, RSV B	94 samples	2023 hr
RV-plus (2017)	LG Chem, Ltd.	NPS, NS, TS	SLAN-96P	~100 min	-	Flu A, Flu B, RSV A, RSV B, OC43, HEV, BoV,	22 samples	5758 hr
RV-16 (2013)	Seegene, Inc.	NPS, NPA, BAL	CFX96	~230 min	-	Flu A, Flu B, RSV A, RSV B, AdV, MPV, BoV, 229E, NL63, OC43, HRV, HEV, PIV (1,2,3,4)	46 samples	6337 hr

Abbreviation: NPS, nasopharyngeal swab; NPA, nasopharyngeal aspirate; BAL, bronchoalveolar lavage; TS, throat swab; OS, oropharyngeal swab; NS, Nasal swab; S, spike protein of SARS-CoV2; E, gene encoding the envelope protein of SARS-CoV-2; RdRp gene, RNA dependent RNA polymerase of SARS-CoV2; N, nucleocapsid protein of SARS-CoV2; Flu A, influenza virus type A; Flu B, influenza virus type B; RSV A, respiratory syncytial virus A; RSV B, respiratory syncytial virus B, HEV, human enterovirus; BoV, bocavirus; HRV, human rhinovirus; PIV (1,2,3,4), parainfluenza virus type 1, 2, 3, 4; TAT, turnaround time.

**Table 2 diagnostics-11-01084-t002:** Comparison of the clinical performance between different assay systems for the detection of SARS-CoV2 and RV in NPS samples.

Assay	Target	SC2FabR Agreement					Sen (%)	Spe (%)	DA (%)
		TP	FP	TN	FN	Total (*n*)			
2019-nCoV	SARS-CoV2	180	0	200	0	380	100	100	100
Standard M	SARS-CoV2	180	0	200	0	380	100	100	100
RP1	Flu A	99	0	391	0	490	100	100	100
	Flu B	91	0	399	0	490	100	100	100
	RSV A/B	74	0	415	1	490	98.7	100	99.8
RV-plus	Flu A	98	2	389	1	490	99.0	99.5	99.4
	Flu B	91	1	398	0	490	100	99.7	99.8
	RSV A/B	69	2	413	6	490	92.0	99.5	98.4

Abbreviations: SARS-CoV2, severe acute respiratory syndrome coronavirus 2; Flu A, influenza virus type A; Flu B, influenza virus type B; RSV A/B, respiratory syncytial virus A/B; TP, true positive; FP, false positive; TN, true negative; FN, false negative; sen, sensitivity; spe, specificity; DA, diagnostic accuracy.

**Table 3 diagnostics-11-01084-t003:** Resolution of discordant sample identification by comparing test assay and RV-16 assay performance.

Sample ID	Discordance Type	SC2FabR Result	RP1 Result	RV-Plus Result	RV-16 Result	Final Determination
A	Flu A	Flu A	Flu A	Neg.	Neg.	Flu A
B		Neg.	Neg.	Flu A	Neg.	Neg.
C		Neg.	Neg.	Flu A	Neg.	Neg.
D	Flu B	Neg.	Neg.	Flu B	Neg.	Neg.
E	RSV	RSV	RSV A	Neg.	RSV A	RSV
F		RSV	Neg.	RSV A	RSV A	RSV
G		RSV	RSV A	Neg.	RSV A	RSV
H		RSV	RSV B	Neg.	Neg.	RSV
I		RSV	RSV B	Neg.	RSV B	RSV
J		RSV	RSV A	Neg.	Neg.	RSV
K		RSV	RSV A	Neg.	Neg.	RSV
L		Neg.	Neg.	RSV A	Neg.	Neg.
M		Neg.	Neg.	RSV A	Neg.	Neg.

Abbreviation: Flu A, influenza virus type A; Flu B, influenza virus type B; RSV A/B, respiratory syncytial virus A/B; Neg., negative.

## Data Availability

All data are available within the article.

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
