# Peer review of "Assay System for Simultaneous Detection of SARS-CoV-2 and Other Respiratory Viruses"

_diagnostics, 2021, doi:10.3390/diagnostics11061084_

Round 1

Reviewer 1 Report

Overall, this manuscript is organized and decently written. SARS-CoV-2 is causing devastating damages to the whole world population and rapid detection/diagnosis is one of the crucial ways to control the pandemic. The objective of this study is simple and clear and the methodology is within expectation. Although the novelty of this study is extremely limited, comparing the efficacy of different detection kits may still provide necessary data/references for the public health/clinical systems from different countries which utilize the related kits. Some major concerns as hereunder would like to be addressed by the authors. 

Major comments: 

1.) Even though the flu and RSV are most commonly found in RV infected patients, the authors should not rule out and underestimate the possibilities of other respiratory viruses co-infection, especially other CoV species as reported previously (eg: DOI: 10.1128/mSphere.00819-20, 10.1002/jmv.25890). Please provide a strong rationale in supporting why these kits are chosen with such detection limitation and only Flu and RSV were investigated. 

2.) The authors mentioned that the clinical symptoms of Flu A and SARS-CoV-2 infections are difficult to be analyzed or distinguished and both flu and RV are commonly found in SARS-CoV-2 patients. However, the aim of the whole study is to evaluate the commercial kits and see which is useful for co-infection detection. Therefore, the accuracy, sensitivity as well as "the variety of RV able to be detected" should be considered and evaluated to prevent missing out possible RVs co-infection in COVID-19 positive cases. Comparatively, the variety of RVs detected by SC2FabR is relatively limited when compared to RP1, RV-plus and RV-16 as mentioned or even other assays from different companies like biomerieux etc. Please suggest if this assay system is still a valid choice for co-infection detection and state in the article with rationale and comparison provided. 

3.) Please state which strains of SARS-CoV-2 are detected in this study. 

4.) Can those selected kits able to detect different strains of SARS-CoV-2 with similar efficacy? Please validate with results or analyses. 

5.) The authors claim the targeted kit is labor and cost-saving, however, I cannot see any interpretation in related areas. Please provide the relevant analyses. 

    Author Response

    Manuscript ID: diagnostics-1238073

    Title: Assay system for simultaneous detection of SARS-CoV-2 and other respiratory viruses

    The authors wish to thank the Reviewers for their helpful comments, and feel that these have contributed substantially to the substance of the manuscript.

    Answers to Reviewers’ comments

    # Reviewer 1

    Comments and Suggestions for Authors

    Overall, this manuscript is organized and decently written. SARS-CoV-2 is causing devastating damages to the whole world population and rapid detection/diagnosis is one of the crucial ways to control the pandemic. The objective of this study is simple and clear and the methodology is within expectation. Although the novelty of this study is extremely limited, comparing the efficacy of different detection kits may still provide necessary data/references for the public health/clinical systems from different countries which utilize the related kits. Some major concerns as hereunder would like to be addressed by the authors. 

    Major comments: 

    1. Even though the flu and RSV are most commonly found in RV infected patients, the authors should not rule out and underestimate the possibilities of other respiratory viruses co-infection, especially other CoV species as reported previously (eg: DOI: 10.1128/mSphere.00819-20, 10.1002/jmv.25890). Please provide a strong rationale in supporting why these kits are chosen with such detection limitation and only Flu and RSV were investigated. 

    A: We agree with the reviewer’s opinion that the possibility of simultaneous infection by other respiratory viruses is also very important and should be studied continuously. However, RSV and Flu exhibit a typical and significant seasonal epidemiology in many countries [1-2]. More specifically, co-infections by respiratory syncytial virus (RSV) and influenza A accounted for the majority of these viral co-infection cases [3]. Patients infected with SARS-CoV-2, Flu, and RSV exhibit overlapping clinical presentations and these are especially fatal for children, elderly, and in patients with comorbidities, such as immunosuppression [4-8]; however, the treatment and management strategies of the infections caused by these viruses are different [4]. Therefore, there is a clinical need for a robust investigation of co-infections in patients with COVID-19 [3].

    It would be extremely difficult to rapidly process such large number of samples if both corona and respiratory viruses have to be tested separately. However, it is expected that in future it will be easy and quick to process large numbers of SARS-CoV-2 and other respiratory viruses simultaneously as well as individually. Therefore, we will definitely consider the reviewer’s comments for our future studies.

    -------------------------------------------------- Reference ---------------------------------------------------------

    [1] Varela FH, Scotta MC, Polese-Bonatto M, et al. Absence of detection of RSV and influenza during the COVID-19 pandemic in a Brazilian cohort: Likely role of lower transmission in the community. J Glob Health. 2021;11:05007. Published 2021 Mar 1. doi:10.7189/jogh.11.05007.

    [2] Kuchar E, Załęski A, Wronowski M, Krankowska D, Podsiadły E, Brodaczewska K, Lewicka A, Lewicki S, Kieda C, Horban A, Kloc M, Kubiak JZ. Children were less frequently infected with SARS-CoV-2 than adults during 2020 COVID-19 pandemic in Warsaw, Poland. Eur J Clin Microbiol Infect Dis. 2021 Mar;40(3):541-547. doi: 10.1007/s10096-020-04038-9. Epub 2020 Sep 28. PMID: 32986153; PMCID: PMC7520378.

    [3] Lansbury L, Lim B, Baskaran V, Lim WS. Co-infections in people with COVID-19: a systematic review and meta-analysis. J Infect. 2020 Aug;81(2):266-275. doi: 10.1016/j.jinf.2020.05.046. Epub 2020 May 27. PMID: 32473235; PMCID: PMC7255350.

    [4] Leung EC, Chow VC, Lee MK, Tang KP, Li DK, Lai RW. Evaluation of the Xpert Xpress SARS-CoV-2/Flu/RSV Assay for Simultaneous Detection of SARS-CoV-2, Influenza A and B Viruses, and Respiratory Syncytial Virus in Nasopharyngeal Specimens. J Clin Microbiol. 2021;59(4):e02965-20. Published 2021 Mar 19. doi:10.1128/JCM.02965-20

    [5] Nair H, Brooks WA, Katz M, Roca A, Berkley JA, Madhi SA, Simmerman JM, Gordon A, Sato M, Howie S, Krishnan A, Ope M, Lindblade KA, Carosone-Link P, Lucero M, Ochieng W, Kamimoto L, Dueger E, Bhat N, Vong S, Theodoratou E, Chittaganpitch M, Chimah O, Balmaseda A, Buchy P, Harris E, Evans V, Katayose M, Gaur B, O'Callaghan-Gordo C, Goswami D, Arvelo W, Venter M, Briese T, Tokarz R, Widdowson M-A, Mounts AW, Breiman RF, Feikin DR, Klugman KP, Olsen SJ, Gessner BD, Wright PF, Rudan I, Broor S, Simões EAF, Campbell H. 2011. Global burden of respiratory infections due to seasonal influenza in young children: asystematic review and meta-analysis. Lancet 378:1917-1930.

    [6] Hall CB, Weinberg GA, Iwane MK, Blumkin AK, Edwards KM, Staat MA, Auinger P, Griffin MR, Poehling KA, Erdman D, Grijalva CG, Zhu Y, Szilagyi P. 2009. The burden of respiratory syncytial virus infection in young children. New Engl J Med 360:588-598

    [7] Thompson WW, Shay DK, Weintraub E, Brammer L, Cox N, Anderson LJ, Fukuda K. 2003. Mortality associated with influenza and respiratory syncytial virus in the United States. JAMA 289:179-186.

    [8] Falsey AR, Hennessey PA, Formica MA, Cox C, Walsh EE. 2005. Respiratory syncytial virus infection in elderly and high-risk adults. New Engl J Med 352:1749-1759.

    1. The authors mentioned that the clinical symptoms of Flu A and SARS-CoV-2 infections are difficult to be analyzed or distinguished and both flu and RV are commonly found in SARS-CoV-2 patients. However, the aim of the whole study is to evaluate the commercial kits and see which is useful for co-infection detection. Therefore, the accuracy, sensitivity as well as "the variety of RV able to be detected" should be considered and evaluated to prevent missing out possible RVs co-infection in COVID-19 positive cases. Comparatively, the variety of RVs detected by SC2FabR is relatively limited when compared to RP1, RV-plus and RV-16 as mentioned or even other assays from different companies like biomerieux etc. Please suggest if this assay system is still a valid choice for co-infection detection and state in the article with rationale and comparison provided.

    A: Thank you for your valuable comment. As you suggested, the purpose of this study was to analyze the performance of a kit used for detecting simultaneous infection. Therefore, it is much more accurate to analyze the accuracy and sensitivity through analyzing the possibility of simultaneous infection of RV that can occur in COVID-19 positive cases, but the number of samples that are positive for simultaneous infection is currently insufficient, so we plan to conduct this research in our future studies.

    However, since this comment is very important, we analyzed the accuracy of simultaneous infection using a few samples, and provided the results in Table S2. We have described the discussion section in the manuscript and marked it in red (Line 275-277).

    Table S2. Eighteen samples used to evaluate the clinical performance of the kit for simultaneous detection of co-infection.

    Sample

    2019-nCoV

    RP1

    SC2FabR

    Comment

    A

    SARS-CoV-2

    Flu A-H1pdm09

    SARS-CoV-2, Flu A

    Match

    B

    SARS-CoV-2

    Flu A-H1pdm09

    SARS-CoV-2, Flu A

    Match

    C

    SARS-CoV-2

    RSV A

    SARS-CoV-2, RSV

    Match

    D

    SARS-CoV-2

    RSV A

    SARS-CoV-2, RSV

    Match

    E

    SARS-CoV-2

    -

    SARS-CoV-2

    Match

    F

    SARS-CoV-2

    -

    SARS-CoV-2

    Match

    G

    SARS-CoV-2

    -

    SARS-CoV-2

    Match

    H

    SARS-CoV-2

    -

    SARS-CoV-2

    Match

    I

    SARS-CoV-2

    -

    SARS-CoV-2

    Match

    J

    SARS-CoV-2

    -

    SARS-CoV-2

    Match

    K

    SARS-CoV-2

    -

    SARS-CoV-2

    Match

    L

    SARS-CoV-2

    -

    SARS-CoV-2

    Match

    M

    SARS-CoV-2

    -

    SARS-CoV-2

    Match

    N

    SARS-CoV-2

    -

    SARS-CoV-2

    Match

    O

    SARS-CoV-2

    -

    SARS-CoV-2

    Match

    P

    SARS-CoV-2

    -

    SARS-CoV-2

    Match

    Q

    SARS-CoV-2

    -

    SARS-CoV-2

    Match

    R

    SARS-CoV-2

    -

    SARS-CoV-2

    Match

    Abbreviations: SARS-CoV-2, severe acute respiratory syndrome-related coronavirus 2; Flu A-H1pdm09, influenza A virus subtype H1pdm09; RSV, respiratory syncytial virus.

    1. Please state which strains of SARS-CoV-2 are detected in this study. 

    A: All samples from the Korea Centers for Disease Control and Prevention (KCDC) and collected from February to September 2020 were confirmed as SARS-CoV-2. During this period, there was no issue regarding the SARS-CoV-2 variant in Korea. Therefore, it was supposed that the strain is used in this study “Wuhan-Hu-1 SARS-CoV-2 (NCBI accession number NC_045512.2)”.

    1. Can those selected kits able to detect different strains of SARS-CoV-2 with similar efficacy? Please validate with results or analyses.

    A: As a result of analyzing the diagnostic efficiency and analysis capability of the kit using different CoV variants (United Kingdom, South Africa, Brazil, USA-California, -New York), very high diagnostic accuracy was obtained. The results are provided in Table S1 and described in the discussion section of the manuscript (Line 269-270).

    Table S1. Eight strains of different lineages of SARS-CoV-2 used to evaluate the analytical performance.

    Company

    Cat No.

    Lineage

    Description

    2019-nCoV

    STANDARD M

    SC2FabR

    Comment

    NCCP

    43381

    B.1.1.7

    United Kingdom

    SARS-CoV-2

    SARS-CoV-2

    SARS-CoV-2

    Match

    43382

    B.1.351

    South Africa

    SARS-CoV-2

    SARS-CoV-2

    SARS-CoV-2

    Match

    43383

    P.2

    Brazil

    SARS-CoV-2

    SARS-CoV-2

    SARS-CoV-2

    Match

    43384

    B.1.427

    USA (CA)

    SARS-CoV-2

    SARS-CoV-2

    SARS-CoV-2

    Match

    43385

    B.1.429

    USA (CA)

    SARS-CoV-2

    SARS-CoV-2

    SARS-CoV-2

    Match

    43386

    B.1.525

    International

    SARS-CoV-2

    SARS-CoV-2

    SARS-CoV-2

    Match

    43387

    B.1.526

    USA (NY)

    SARS-CoV-2

    SARS-CoV-2

    SARS-CoV-2

    Match

    Twist Bio

    79683

    P.1

    Brazil

    SARS-CoV-2

    SARS-CoV-2

    SARS-CoV-2

    Match

    Strain description, cited by the PANGO lineages from which the SARS-CoV-2 strain was acquired, is shown. Abbreviations: SARS-CoV-2, severe acute respiratory syndrome-related coronavirus 2; NCCP, National Culture Collection for pathogens.

    1. The authors claim the targeted kit is labor and cost-saving, however, I cannot see any interpretation in related areas. Please provide the relevant analyses.

    A: Thank you for your valuable comment. During the COVID-19 pandemic, approximately 76,000 samples were screened daily for disease diagnosis in Korea [1]. To compare the efficiency of the SC2FabR, we estimated the turnaround time (TAT) using a combination of kits (2019-nCoV & RP1, 2019-nCoV& RV-plus, STANDARD M & RP1, and STADMARD M & RV-plus) in this study. As a result, it was confirmed that the number of tubes used for analysis and TAT were significantly reduced upon conducting simultaneous analysis of SC2FabR (see below). As you suggested, we have added the capacity of 96 well-plate and turnaround time in Table 1 and marks it in red.

    -------------------------------------------------- Reference ---------------------------------------------------------

    [1] Korea Centers for Disease Control and Prevention. Updates on COVID-19 in Republic of Korea. Available online: http://ncov.mohw.go.kr/tcmBoardView.do?brdId=3&brdGubun=31 &dataGubun=&ncvContSeq=5484&contSeq=5484&board_id=311&gubun=ALL (accessed on 1 June).

    Reviewer 2 Report

    The study of Lim et al entitled "Assay system for simultaneous detection of SARS-CoV-2 and other respiratory viruses." compared the analytical performance of the SC2FabR platform with four other commercial analytical platforms. This is an important study which can provide reference to clinical diagnostic laboratory under the current COVID-19 pandemic. The study design and result presentation are of high quality.

    Comments:

    1. The authors may include the price comparison of each system and each assay in Table 1.
    2. Refer to Table 1 SC2FabR is also optimised with saliva samples, with are non-invasive when comparing with NPS, which is indeed an advantage over other system, maybe the authors can comment on this in the manuscript as well. 

    Author Response

    Manuscript ID: diagnostics-1238073

    Title: Assay system for simultaneous detection of SARS-CoV-2 and other respiratory viruses

    The authors wish to thank the Reviewers for their helpful comments, and feel that these have contributed substantially to the substance of the manuscript.

    Answers to Reviewers’ comments

    # Reviewer 2

    Comments and Suggestions for Authors

    The study of Lim et al entitled "Assay system for simultaneous detection of SARS-CoV-2 and other respiratory viruses." compared the analytical performance of the SC2FabR platform with four other commercial analytical platforms. This is an important study which can provide reference to clinical diagnostic laboratory under the current COVID-19 pandemic. The study design and result presentation are of high quality.

    Comments:

    1. The authors may include the price comparison of each system and each assay in Table 1

    A: Thank you for your valuable comment. Your opinion is also an important part, but we couldn’t add the price information. Because the variability is multifactorial, including nation and agency etc., it would be difficult to correctly compare the price of each system and each assay. However, based on your valuable comment, we have listed the capacity of 96 well-plate and turnaround time (TAT) of each system and each assay instead of pricing information in Table 1.

    1. Refer to Table 1 SC2FabR is also optimized with saliva samples, with are non-invasive when comparing with NPS, which is indeed an advantage over other system, maybe the authors can comment on this in the manuscript as well.

    A: Thank you for your valuable comment. As you suggested, we have added the non-invasive advantage of using saliva samples in discussion and marks it in red (Line 289-291).

    Round 2

    Reviewer 1 Report

    Overall, the authors have addressed the comments adequately.

    One minor issue would like to address concerning the SARS-CoV-2 variants, the authors should mention the assumption of the detected variants belong to the Wuhan-Hu-1 viral strain in the manuscript since it is an important message for epidemiologists to trace and study the evolutionary pathway of SARS-CoV-2 as well as the clinicians to determine if these kits are valid choices for their detection at current pandemic stage. 

    Author Response

    Manuscript ID: diagnostics-1238073

    Title: Assay system for simultaneous detection of SARS-CoV-2 and other respiratory viruses

    The authors wish to thank the Reviewers for their helpful comments, and feel that these have contributed substantially to the substance of the manuscript.

    Answers to Reviewers’ comments

    # Reviewer 1

    Comments and Suggestions for Authors

    Overall, the authors have addressed the comments adequately.

    Minor comments: 

    1. One minor issue would like to address concerning the SARS-CoV-2 variants, the authors should mention the assumption of the detected variants belong to the Wuhan-Hu-1 viral strain in the manuscript since it is an important message for epidemiologists to trace and study the evolutionary pathway of SARS-CoV-2 as well as the clinicians to determine if these kits are valid choices for their detection at current pandemic stage.

    A: Thank you for your valuable comment. As you suggested, we mentioned the strain used in this study as the Wuhan-Hu-1 viral strain (NCBI accession number NC_045512.2) in the discussion section of the manuscript and tracked it (Line 268-270).